# Essential gene complement of *Planctopirus limnophila* from the bacterial phylum *Planctomycetes*

Elena Rivas-Marin [1,4] ✉, David Moyano-Palazuelo [1,4], Valentina Henriques [1], Enrique Merino[2] & Damien P. Devos [1,3] ✉

*Planctopirus limnophila* belongs to the bacterial phylum *Planctomycetes*, a relatively understudied lineage with remarkable cell biology features. Here, we report a genome-wide analysis of essential gene content in *P. limnophila*. We show that certain genes involved in peptidoglycan synthesis or cell division, which are essential in most other studied bacteria, are not essential for growth under laboratory conditions in this species. We identify essential genes likely involved in lipopolysaccharide biosynthesis, consistent with the view of Planctomycetes as diderm bacteria, and highlight other essential genes of unknown functions. Furthermore, we explore potential stages of evolution of the essential gene repertoire in *Planctomycetes* and the related phyla *Verrucomicrobia* and *Chlamydiae*. Our results provide insights into the divergent molecular and cellular biology of *Planctomycetes*.

Most of our knowledge about molecular and cellular microbiology is derived from a few branches in the bacterial tree, *Alpha* and *Gammaproteobacteria*, *Firmicutes,* and *Bacilli*. However, next-generation sequencing techniques have revealed the gap existing between the organisms that can be grown in the lab and the diversity of the ones found outside of it[1–7], evidencing the enormous number of exciting fundamentals in biology to be discovered.

All this biodiversity is genome encoded and deciphering this novel biology requires obtaining and understanding genomic information. However, in each genome, the number of genetic elements of unknown function is important. Usually, for model organisms, between half and three-quarters of their proteome can be functionally annotated[2,3]. This situation differs for the vast majority of non-model organisms, where often the majority of the components encoded in their genomes cannot be functionally assigned[4].

Function definition and assignment to proteins are complex problems by themselves and huge efforts have been devoted to them[8]. One of the most basic and fundamental aspects of protein function is linked to the essentiality of its encoding gene. A gene is defined as essential if its presence in a genome is required for growth, which is dependent on the environmental conditions[9]. There are many motivations to define lists of genes that are essential for survival, or important for optimal growth rates, including the determination of pathogenicity, antibiotic resistance, identification of non-coding RNAs (ncRNAs), as well as drug and vaccine targets.

Transposon-based approaches have been applied over the last 15 years to hundreds of different strains, mainly bacteria, including the bacterium *E. coli*[10]; human, animal, or plant pathogens[11–16], as well as bacteria with potential biotechnological value[17], deciphering novel biology and many leads for gene function[18]. The transposon-directed insertion site sequencing (TraDIS)[19] method is one such approach that combines exhaustive Tn*5* transposon mutagenesis with next-generation sequencing to determine the essentiality or fitness contribution of each genetic feature in a genome simultaneously[18]. Additionally, saturated libraries have been used to establish essential intergenic regions, ncRNAs, and regulatory elements[20] or conditional essential genes under chosen situations[21].

*Planctomycetes*, belonging to the bacterial *Planctomycetes-Verrucomicrobia-Chlamydiae* (PVC) superphylum[22], is one of these phyla containing divergent biology[1,23–28]. Among others, Planctomycetes

[1]Centro Andaluz de Biología del Desarrollo, CSIC, Universidad Pablo de Olavide, Sevilla, Spain. [2]Instituto de Biotecnología, Universidad Nacional Autónoma de México, Cuernavaca, Morelos, México. [3]Present address: Institut Pasteur de Lille, Centre d'Infection et d'Immunité de Lille, University of Lille, Lille, France. [4]These authors contributed equally: Elena Rivas-Marin, David Moyano-Palazuelo. ✉e-mail: erivmar@upo.es; damienpdevos@gmail.com

have a developed endomembrane system and divide without FtsZ, the otherwise universal division protein in *Bacteria*[27,29]. These and other features have recently raised interest, promoting an important effort in sampling, which translates into the exponential growth of the sequence data, including from non-cultured strains[1,30,31]. Planctomycetes have large genomes for bacteria ranging from 4.5 to 12 Mb in size, containing more sequences of unknown than known function[32].

In order to contribute to deciphering planctomycetal molecular and cell biology, we have applied TraDIS technology to establish the essentiality of every genetic element of *Planctopirus limnophila*. We categorized its 4327 genes, including ncRNAs, as either essential or non-essential, under standard laboratory conditions. We defined a reference list of the essential genes in this species, serving as a scaffold for Planctomycetes gene function determination. Furthermore, we identified clear cases in which only a fraction of a gene lacks transposons, suggesting that only a fragment of the protein is essential, previously defined as "domain essentiality"[10]. As all members of the *Planctomycetes* divide using an unknown molecular machinery, we focused on genes related to cell division. We confirm that most of the genes associated with division, which are usually essential in other bacteria, are dispensable in *P. limnophila*. Furthermore, we reveal that most of the genes involved in peptidoglycan (PG) synthesis are also not essential in this species, addressing a long-standing controversy in the phylum[33,34]. Eventually, phyletic profiling linked to essentiality revealed gene evolution related to the divergence of these species. In this work, we establish a genomic tool to be used as a reference for further analyses of this strain or other bacteria within the phylum or superphylum.

## Results and discussion

### Saturated transposition in *Planctopirus limnophila* genome

A TraDIS method was applied to *P. limnophila* in order to obtain a transposon mutant library[10,35]. A complex consisting of a mini-Tn*5* transposon bearing a kanamycin resistance cassette together with a transposase was electroporated into competent cells and grown on a selective medium. Individual colonies were collected to construct the library, estimated to be approximately 1.1 million mutants. The pooled library was sequenced before storage or outgrowth to map the location of the insertions. Sequencing data were obtained from extracts of the transposon library, resulting in 8,132,446 sequence reads. After the removal of short reads, poor-quality data, and fragments that did not contain the transposon sequence, 2,863,686 reads were mapped to the *P. limnophila* genome. This resulted in 505,437 unique insertions on the main chromosome and 2734 in the plasmid (Fig. 1; Sup. Data 1).

The insertion sites cover the entire genome length and are evenly distributed; no biases in the transposition events could be detected. 415,995 insertion events mapped in coding sequences (CDS) of the chromosome, leaving 89,442 events in the intergenic regions. In the plasmid, 2516 insertions were found in CDS and 218 in intergenic regions. The high density of unique insertion sites resulted in an average of one insertion every 11 bp in the chromosome and every 14 bp in the plasmid.

### Gene essentiality determination

We first followed a statistical method defined previously[10] to assign a score of essentiality to each gene in the genome. Briefly, the numbers of unique insertion sites per CDS are quantified and normalized by gene length. As expected, the frequency distribution of the insertion index scores was bimodal (Fig. S1). Genes associated with the left mode were defined as essential, and the ones associated with the right mode as non-essential, while genes in between were deemed as unclear. Two distribution models, gamma and exponential, were fitted to the frequency distribution, and the probability of each gene belonging to one or the other was defined. The ratio of these values defined the log-likelihood score. A gene was classified as essential if its log-likelihood score was less than $\log_2(12)$, therefore, if it was 12 times more likely to belong to the essential mode than to the non-essential one.

Using this technique, from the 4327 genes (including 4258 CDS, 61 tRNA genes, 2 16S rRNA, one 23S rRNA and one 5S rRNA) plus 46 pseudogenes in *P. limnophila*, we identified 739 genes as essential, 3481 as non-essential and 153 as unclear (17%, 80%, and 3%, respectively: Sup. Data 2). The list of essential genes includes the usual

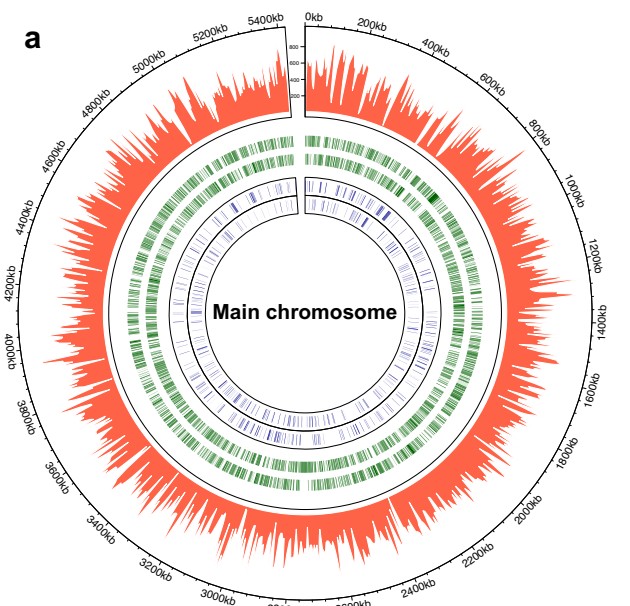
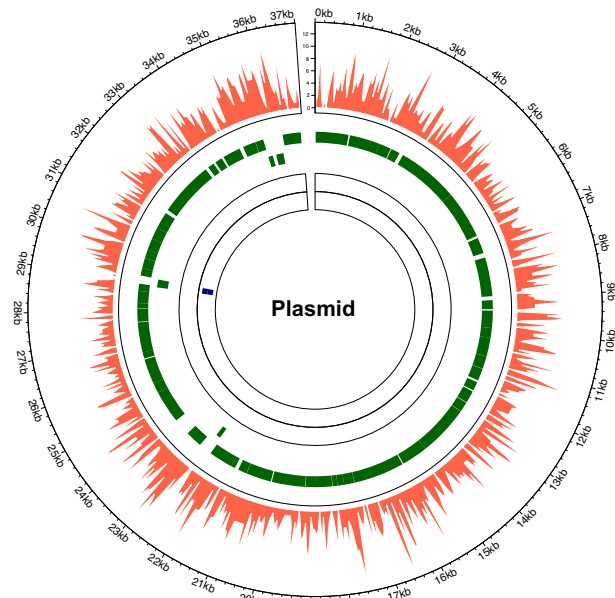

**Fig. 1 | Genome and plasmid-wide transposon insertion sites of *P. limnophila* transposon library.** Frequency and location of transposon junction sequences from a mini-Tn*5* transposon library mapped to the *P. limnophila* genome. The outermost track marks the *P. limnophila* genome in base pairs starting at the annotation origin. The next inner tracks belong to the frequency and location of insertion sequences in the *P. limnophila* genome (red). The four innermost circles correspond to sense and anti-sense CDS, respectively (green), followed by two inner tracks depicting the essential genes identified by TraDIS on the sense and anti-sense strands, respectively (blue). **a** and **b** represent the main chromosome and the plasmid.

suspects, such as the ones coding for the four subunits of the RNA polymerase core, the housekeeping sigma factor, and almost all the 54 ribosomal proteins. Only one gene from the plasmid was essential; it codes for a protein of unknown function. We also identified two genes annotated as pseudogenes (Plim_2061 and Plim_4186) with a low insertion index score. According to our criteria, those are essential and thus bonafide genes. Modifying the parameters of bin widths and troughs of the distributions of our statistical analysis, or the cut-offs, did not result in significant modifications to the list of essential genes; only some genes were added or removed, showcasing the robustness of our analysis (Sup. Data 3). Manual inspection of the unclear genes reveals that a significant proportion is explained by biased insertions to only a portion of the gene, revealing that only a domain of the protein is essential[10]. We thus defined a second method based on a sliding window, where a gene is defined as essential if at least a fragment of 300 bp of the CDS is free of insertions (see "Methods" section). This additional step detected 28 additional unclear genes and 9 non-essential as "domain essential" genes, raising the total of essential genes to 776 (17.8%) (Sup. Data 2 and 4, Fig. S2). The unclear genes include Plim_0500, coding for the 50S ribosomal protein L17, and Plim_3650, coding for the GlmU protein (Fig. 2). GlmU is a bifunctional enzyme that catalyzes sequential steps in the biosynthesis of UDP-N-acetyl-D-glucosamine (UDP-GlcNAc), an essential precursor of the cell wall components PG and lipopolysaccharide. The enzymatic activities of GlmU are present in two independently folding and functional domains, MobA-like NTP transferase domain and Bacterial transferase hexapeptide domain; only the first of these is essential in *P. limnophila*. As for the genes previously reported as non-essential, only 9 genes exhibit domain essentiality, most of which have unknown functions. Only Plim_2324 and Plim_3011 have known functions, annotated as adenylyl-transferase (Glutamate–ammonia-ligase) and DNA polymerase I, respectively. In the latter case, only the domains associated with exonuclease activity appear to be essential.

To link essentiality and function, we used functional labels of Cluster of Ortholog Groups of proteins (COG)[36]. Of the 4258 proteins in the *P. limnophila* proteome, 1248 (29%) proteins were not assigned to a COG, and 644 (15%) were annotated as class S (Unknown function). Excluding the proteins annotated as class S, 2072 had a single COG functional label and 294 had more than one. As for non-essential genes, 45% of them were not annotated or of unknown function. In contrast, the majority of the essential genes annotated belong to the classes "Translation, ribosomal structure and biogenesis (J)", "Energy production and conversion (C)", "Coenzyme transport and metabolism (H)", followed by genes of "Function unknown (S)" (Fig. 3). Interestingly, 151 essential CDS (of 748, ~20%) are of unknown function (135 hypothetical proteins and 16 domain of unknown function or DUF), emphasizing the relevance of the novel biology in this organism.

There are, however, a large number of paralogies of these enzymes, which might explain their non-essential nature in *P. limnophila*. We assess this possibility in the section "Paralogs and essentiality".

**Comparative essentiality**

The percentage of essential genes in the *P. limnophila* genomes (17,9%) is in the upper range when compared to other bacteria, e.g., 14.4% for *Brucella abortus*[37], 13.4% for *Brevundimonas subvibrioides*[38], 12.19% for *Caulobacter crescentus*[39] and 8.3% for *E. coli*[10]. However, a more precise comparison between organisms should consider more variables, including the lifestyles, the conditions, and the composition of the medium used to define this essentiality.

In some cases, a low insertion index for a determined gene that categorized it as essential could be due to the high level of genome condensation or its location close to the replication terminus, which prevents transposition events[40,41]. *Planctomycetia* genomic DNA is known to be highly condensed[42], most likely associated with DNA-binding proteins, which might imply that a portion of the genes appearing as essential is in fact, not accessible to the transposase.

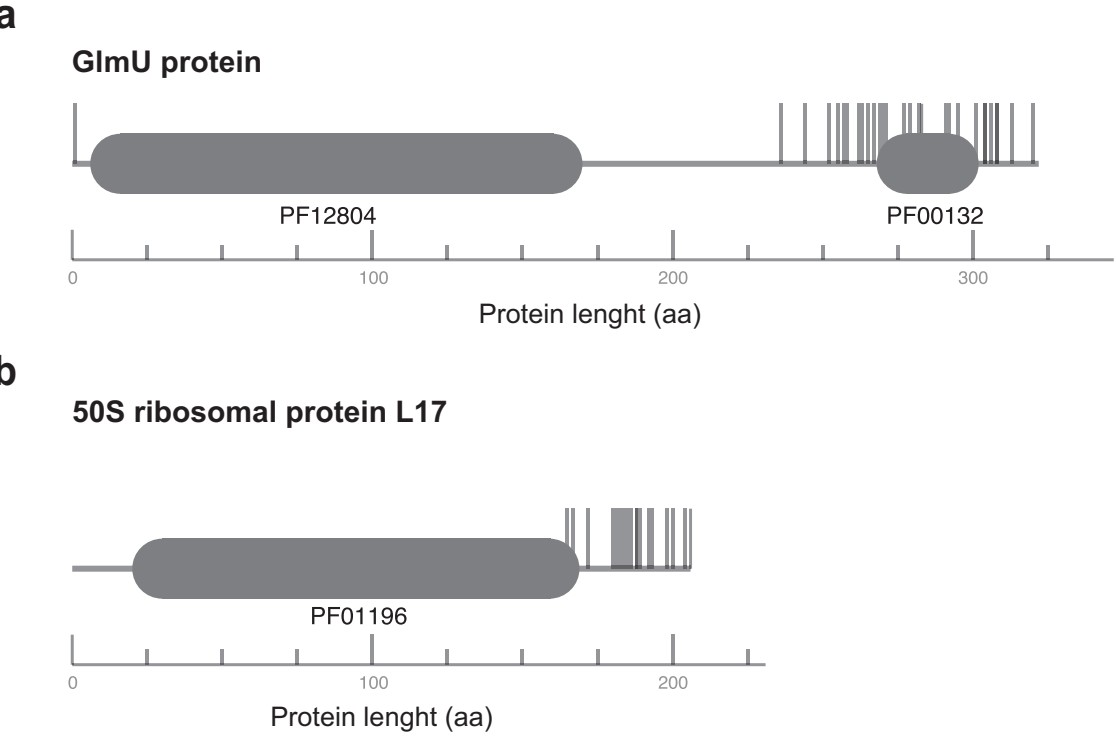

**Fig. 2 | Examples of 'domain essential' genes in *P. limnophila*.** Vertical lines represent transposon insertion sites. Pfam protein domains are represented in gray with the ID shown below. The horizontal axis represents the protein length. **a** GlmU protein (Plim_3650) with the domains MobA-like NTP transferase domain (PF12804) and Bacterial transferase hexapeptide (PF00132), **b** 50S ribosomal protein L17 (Plim_0500) with the domain Ribosomal protein L17 (PF01196).

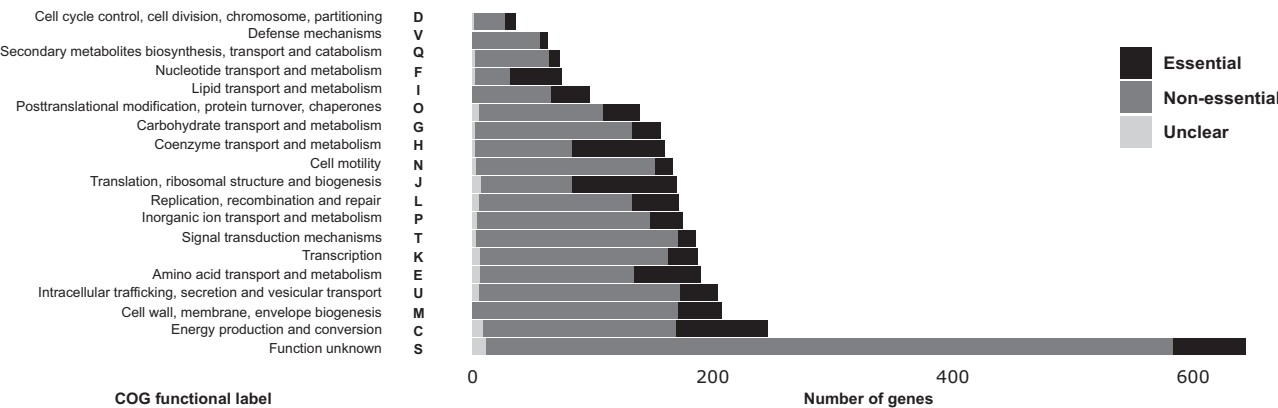

**Fig. 3 | Distribution of COG functional labels grouped by essentiality.** The *x*-axis represents the number of genes annotated with a COG functional label, according to proGenomes[66]. In each bar, the number of essential genes is represented in black, non-essential genes in gray, and unclear genes in light gray. Source data are provided as a Source Data file.

To contrast these numbers, we compared the *P. limnophila* results with the ones available for *E. coli*[10], the workhorse model of microorganisms. As the *E. coli* data exclusively contain CDS, we considered only the 4258 *P. limnophila* CDS, of which 711 are essential, 3404 are non-essential, and 143 are unclear. We did not include the 'domain essential' genes in the list, as this method was not applied to the *E. coli* dataset.

We searched for orthologs between *P. limnophila* and *E. coli* proteins by bidirectional best-hit, discarding proteins with paralogies that are difficult to resolve. There are 440 such direct orthologs reflecting the phylogenetic separation between the two species (Sup. Data 5), 312 (70.9%) of which agree on essentiality. Of these, 122 (39.1%) were essential, 189 (60.6%) were not, and one was unclear. The essential genes in both organisms define the invariable essential gene set, primarily related to central metabolism (Sup. Data 5). These mainly include genes involved in DNA maintenance and repair, ribosomal proteins, the major subunits of the RNA polymerase, the SecY subunit of the translocase, elements of the lipid A synthase, and aminoacyl-tRNA synthetases. On the other side, different essentialities found for 128 genes could reflect different biology. We found 19 genes defined as essential in *E. coli* that are not essential in *P. limnophila*. This group includes most of the genes coding for the PG synthesis enzymes. We also found 73 genes that were not essential in *E. coli* but essential in *P. limnophila*. Those genes are mostly related to amino acid, nucleotide, and coenzyme transport, and metabolism. Those 73 genes also include one coding for a protein of unknown function, Plim_0829, containing a domain of unknown function (DUF205).

In addition to reflect differences in the biology of these organisms, parts of these discrepancies might be attributable to the differences in medium composition. TraDIS might help define a specific minimal medium for any organism.

## Conservation

To further characterize the novel biology of Planctomycetes, we contrasted the essentiality and conservation of the *P. limnophila* proteins

in the *Planctomycetes* phylum, the PVC superphylum, and the prokaryotic domains. A total of 4655 organisms collected in the Kyoto Encyclopedia of Genes and Genomes (KEGG) database[43] were considered, of which 4365 bacteria (including 79 PVC: 45 Planctomycetes, 17 Verrucomicrobia, and 17 Chlamydiae) and 290 archaea (Table 1). Among the universal proteins conserved within prokaryotes, we found 49 ribosomal proteins, 21 aminoacyl-tRNA synthetases, MreB (Plim_2620, annotated as "cell shape determining protein, MreB/Mrl family"), and FtsK (Plim_2063) (Sup. Data 6). Some regions of the *P. limnophila* genome contain genes that are both conserved in many organisms and essential. This is the case of the *locus* containing the eight ATP synthase subunits (Plim_0170 to Plim_0177). Interestingly, and regardless of their relevance in the majority of the organisms, the last gene Plim_0177, coding for the epsilon chain, is reported as non-essential in our study, and the three first subunits are specific to Planctomycetes (Fig. 4a). Another example is the ribosomal genomic fragment (Plim_0472 to Plim_0500), that corresponds to the biggest putative operon in the *P. limnophila* genome with 29 genes, according to Operon-mapper[44]. Ribosomal proteins are highly conserved in prokaryotes, including *Planctomycetes* and PVC, and all are essential, with one exception, Plim_0485, coding for the subunit L29. This genomic fragment also contains the genes coding for the translation elongation factor G, the translocase subunit SecY, and the DNA-directed RNA polymerase alpha and beta' subunits (Plim_0475, Plim_0496, Plim_0499, and Plim_0472, respectively).

The locus of genes from Plim_0410 to Plim_0401 contains one of the largest putative operons in *P. limnophila*, composed of eight genes (Plim_0409 to Plim_0402). Six of its proteins of unknown function are restricted to Planctomycetes (Plim_0406 to Plim_0401; Fig. 4b). The first four genes (Plim_0410 to Plim_0407) are found in most bacteria and annotated as Type II secretion protein. This suggests that the functions of these proteins are related to secretion, although six of them appear to be related to *Planctomycetes* specificities.

From the 4258 proteins in the *P. limnophila* proteome, 1519 do not have known homologs outside the PVC superphylum of which 1477 are specific to *Planctomycetes*. A considerable number of essential proteins of unknown function, 108, are specific to the phylum, making them attractive targets for further studies. The majority of the genes specific to *Planctomycetes* were not annotated or annotated as DUF (Table 1). Together with the number of proteins specific to the phylum, our results emphasize the unique biology of this group and provide leads toward identifying the actors behind the divergent biology of *Planctomycetes*.

Due to the phylogenetic divergence between this organism and most of the organisms considered as models, we sought to use a complementary method based on ortholog groups. To investigate the

**Table 1 | Conservation of *P. limnophila* genes at different taxonomic levels**

| Shared with | Essential | Non-essential | Unclear | Total |
|---|---|---|---|---|
| Prokaryotes | 612 | 2065 | 62 | 2739 |
| PVC | 136 | 1330 | 53 | 1519 |
| Planctomycetes | 136 | 1289 | 52 | 1477 |
| Planctomycetes (hypothetical protein/DUF) | 108 | 1128 | 45 | 1281 |

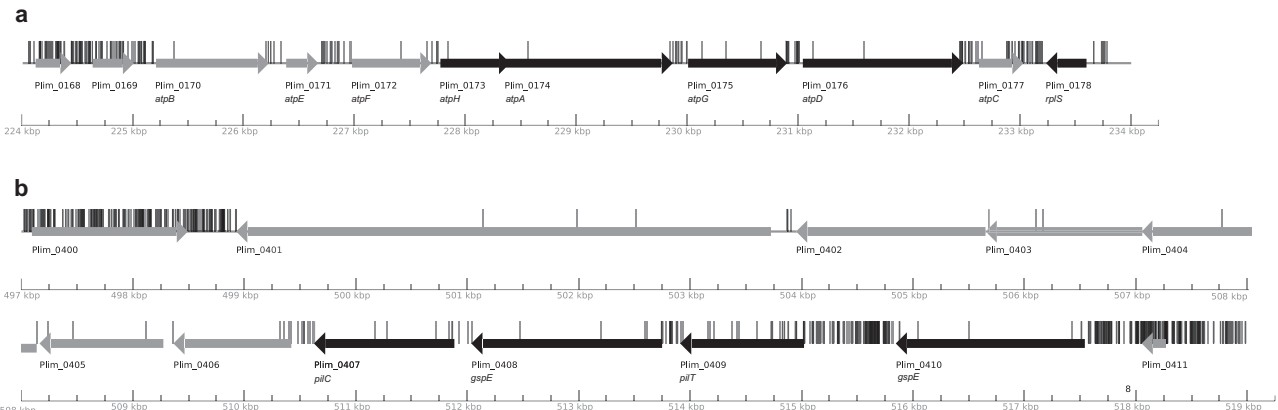

**Fig. 4 | Schematic representation of transposon insertions.** Transposon insertion sites in the **a** genomic fragment surrounding the genes coding for subunits of the ATP synthase (Plim_0168-0178) and **b** the fragment surrounding the genes coding protein related to secretion (Plim_0401-Plim_0410). Genes are represented with an arrow, and vertical lines represent transposon insertion sites. Different colors show differences in gene conservation. Genes colored in black are highly conserved in all prokaryotes, while genes in gray are not.

broader phyletic distribution of the *P. limnophila* essential genes, we assigned them to the EggNOG orthologs groups to map the presence and absence of putative essential gene orthologs from 5277 complete genomes grouped on 50 classes, representing major clades in all 3 domains[45,46]. A graphical representation of the *P. limnophila* essential genes shared in other genomes in a set of hierarchical clusters based on Euclidean distance provides a visual display of the transitions in the evolution of the *P. limnophila* essential genome (Fig. S3). The biggest set of essential proteins is distributed universally in bacteria, including some that are also found in archaea and/or eukaryotes (Group A Fig. 5a). The next set in size is semi-distributed in prokaryotes, most prevalent in *Gracillicutes* (Group B). The smallest group is composed of genes restricted to almost only *Planctomycetes* (Group C). This group contains the YTV domain protein, one of the few that has been assayed experimentally[47]. The YTV domain is restricted to the class *Planctomycetia* and is suggested to be involved in the regulation of cell wall rigidity due to its high content of cysteine. This approach revealed the divergent biology of *Planctomycetes* due to novel genes, including the novel division mode (Fig. 5a). The same process based on the essential genes from *E. coli* reveals genes that have been lost, such as *ftsZ* (Fig. 5b, Green box; Fig. S4), *ftsA* and *ftsB*, coding for known partners of FtsZ, which have been lost in a similar pattern (Fig. 5b, blue box). This approach discriminates between the classes *Planctomycetia* and *Phycisphaera*. The presence of most of the division-associated genes, e.g., *ftsW*, is scarce in the class *Planctomycetia*, while conserved in *Phycisphaera* (Fig. 5b, Blue box).

The genes related to lipopolysaccharides (LPS) synthesis are mostly lost in *Terrabacteria* (Fig. 5b, Pink box), while the conservation in *Planctomycetes* support their diderm cell type[29]. The genes that are mostly lost in *Phycisphaera*, while conserved in *Planctomycetia*, are related to specific metabolism, highlighting the divergent biology of *Phycisphaera* (Fig. 5b, Orange box) within the *Planctomycetes*. Thus, this analysis is robust enough to reveal gene gains or losses behind the divergent biology of various organisms.

## Paralogs and essentiality

In order to evaluate the influence of copy number on the essential character of a group of genes, we clustered the *P. limnophila* genes in 328 paralog groups, containing 1075 genes, 160 of which are essential, 890 are not and 25 are unclear (Sup. Data 7). The biggest group has 68 paralogs, those proteins are of unknown function and contain the DUF1559, a large family of paralogous proteins apparently restricted to Planctomycetes (source InterPro). The high number of paralogs suggests that the function of this DUF is important for these bacteria.

In fact, two of these proteins are essential in our analysis. Of the 328 paralog groups, 33 contain only essential proteins. This includes a group composed of five proteins annotated as '3-oxoacyl-(acyl-carrier-protein) synthase 2' or 'Beta-ketoacyl synthase'. Another group, annotated as chaperonin GroEL, contains three essential proteins. The remaining 31 groups only contain two proteins whose functions are mostly related to the central maintenance of the cell, such as DNA topoisomerase subunits, NADH dehydrogenase, and chromosome segregation, among others. 208 groups contain only non-essential genes, including six groups with more than ten paralogs. One additional group is composed of two unclear proteins without functional annotation. There are 86 groups of paralogs containing a mixture of essential and non-essential genes, most of them containing between two and four proteins. Twelve of these groups encode proteins of unknown function. The combination of essential and non-essential paralogs might be explained by the complementation of a mutant by at least one of the other copies of this gene. Further studies are requested to clarify this point, e.g., multiple simultaneous mutants.

## Peptidoglycan and cell division: planctomycetes do it differently

For a long time, controversy surrounded the presence of PG in *Planctomycetes*[1,27,28]. Indeed, early on, they were described as lacking this otherwise almost universal feature of the bacterial cell wall, similar to *Chlamydiae*[27]. This controversy was apparently solved when PG was detected in the classes *Planctomycetia* and *Candidatus* Brocadia (anammox Plantomycetes)[48,49], as well as in *Chlamydiae*[50], although in the latter, it is tightly regulated in time and space.

Proteins encoded by the *GlmSMU* genes convert fructose-6-phosphate into N-acetylglucosamine (UDP-GlcNAc), entering the PG synthesis pathway. Those genes are essential, including, *glmU*, a clear case of domain essential gene (Fig. 2a). The genes coding for these proteins show a pattern of losses in most members of the *Planctomycetia* (Fig. 5b; Blue box). All the genes coding for the enzymes realizing subsequent steps of the pathway (*murABCDEFGJ*) are not essential in our screening, contrasting with their essential character in *E. coli* and most bacteria. This is also the case for *murB*, which is not essential despite being very conserved in *Planctomycetia*, and *murE* which is semi-conserved in this class.

MraY transfers the UDP-MurNAc-pentapeptide onto the lipid carrier undecaprenyl phosphate, yielding lipid I[51]. MurG then adds a UDP-GlcNAc subunit to the lipid I attached to the membrane producing lipid II. MurJ is involved in flipping the lipid II to the periplasm and finalizing the cytoplasmic steps of the PG synthesis[52]. Most of these genes are essential in *E. coli* and other bacteria, but none of them are

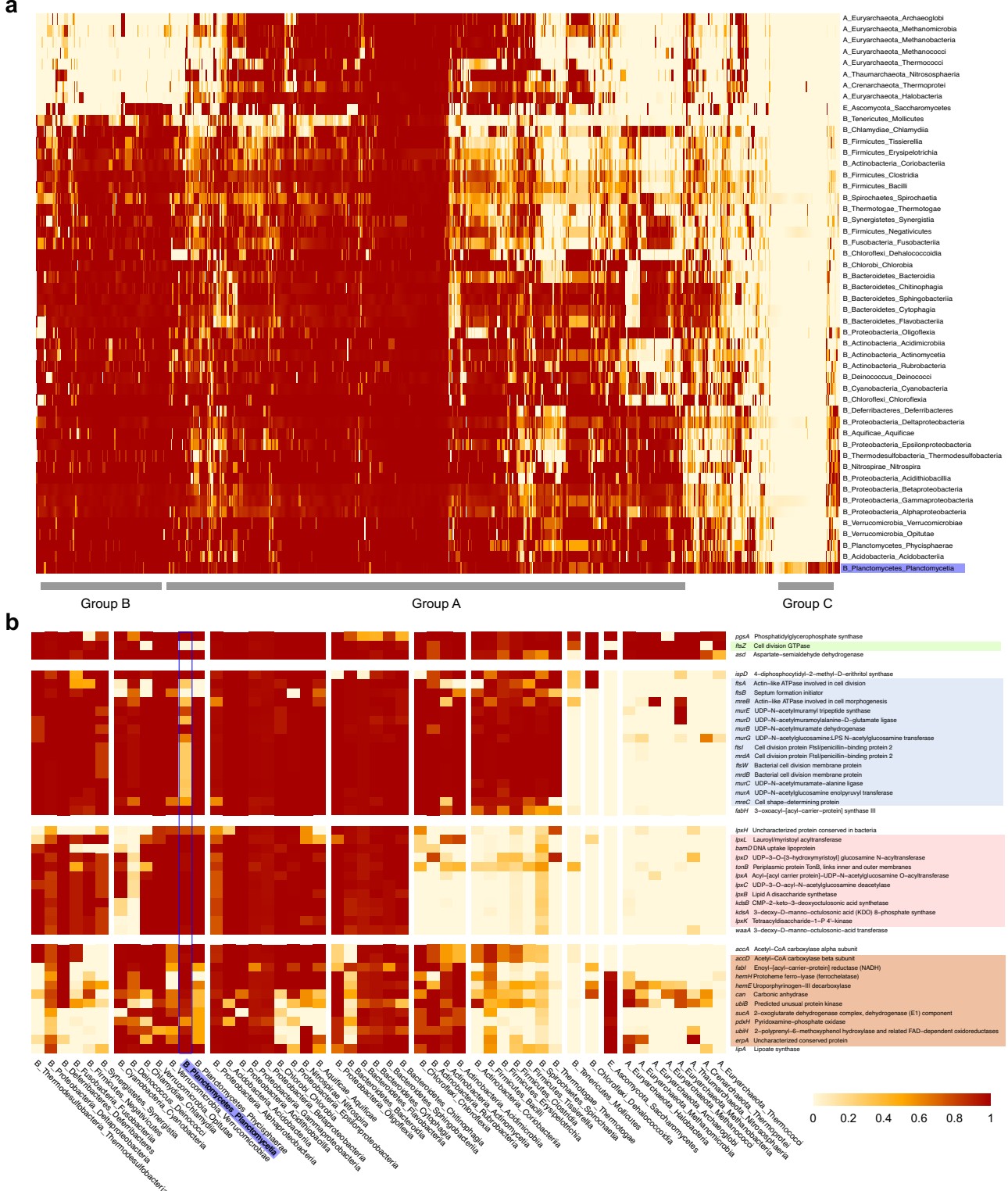

**Fig. 5 | Phyletic patterns of *P. limnophila* and *E. coli* essential genes. a** EggNOG orthologs group (NOG) heatmap according to the EggNOG database annotations for the *P. limnophila* essential gene set. Each row represents one of the fifty selected classes. Each column represents the essential genes discovered in this study with a NOG. **b** Selection of proteins of interest (in green, blue, pink and orange boxes) from the NOG heatmap based on EggNOG-mapper annotations for the *E. coli* essential protein set. Each row represents the essential genes reported by Goodall et al.[10] with a NOG. GenBank ID of the gene encoding this protein and the gene name, followed by the NOG with its functional category and function (right). Each column represents one of the fifty selected classes. The heatmap color reflects the degree of distribution of a particular NOG in each class, with red representing complete distribution in all selected organisms of that class and light yellow representing absence in all selected organisms of that class.

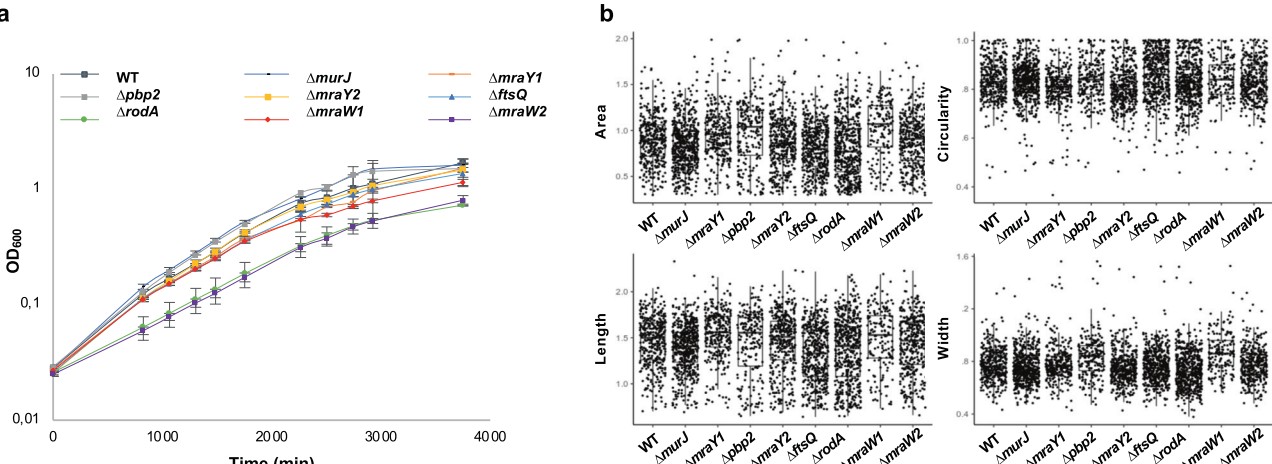

**Fig. 6 | Cell division *P. limnophila* mutants. a** Growth curve of *P. limnophila* deletion mutants using OD measurements represented in semilogarithmic scale. Data points are mean ± standard deviation (S.D.) for *n* = 3 biological replicates. **b** Morphological measurements (area, circularity, length and width) of the mutant cells in μm (*n* = 371, 517, 214, 539, 390, 287, 371, 165, 469 for wild type, Δ*rodA*, Δ*pbp2*, Δ*murJ*, Δ*mraY2*, Δ*mraY1*, Δ*mraW2*, Δ*mraW1*, Δ*ftsQ*, respectively). Box plots

of the corresponding measurements where the median, two hinges and two whiskers, and all outlying points are displayed. The lower and upper hinges correspond to the first and third quartiles. Whisker extends to the smallest and largest values, no further than 1.5 * distance between the first and third quartiles from the hinge. Source data are provided as a Source Data file.

essential in our analyses. These results confirm that, despite the likely presence of PG in Planctomycetes[48,49], its synthesis, in this strain, and probably in this phylum, presents significant differences with respect to the way in which it is canonically synthesized in model organisms. To keep exploring the divergent biology of Planctomycetes, we further investigated those genes involved in cell division. Some Planctomycetes divide by binary fission (the *Cand.* Brocadia and class *Phycisphaerae*), while, unlike most other bacteria, others divide by asymmetric division (class *Planctomycetia*)[53]. However, all Planctomycetes have lost the *ftsZ* gene and other genes show a punctuated pattern of presence in genomic profiling[53]. We had previously shown that genes that are otherwise reported as essential in other species and that are almost ubiquitous in bacteria (e.g., *ftsI*, *ftsW*, and *mreB*) are not essential in Planctomycetes[23]. Here, we first confirm the lack of essentiality previously reported for some of the genes from the *dcw* (*division and cell wall*) cluster, and we further extend it to other genes involved in division and PG synthesis. This is the case for the genes *mraY1*, *mraY2*, *mraW1*, *mraW2*, *rodA*, *murJ*, *pbp2* and *ftsQ*. In order to confirm this result, we constructed single deletion mutants for these genes, deleting the whole length of their CDS (Fig. S5). None of these genes were essential and we were able to generate mutants. In contrast, *ftsK* is essential in our screen, in agreement with targeted deletion experiments[23]. Thus, division and PG synthesis are different in Planctomycetes, as previously suggested[47,54]. Growth curves of the mutants did not differ significantly from the wild type in contrast to the expectation (Fig. 6a). Similarly, by measuring different cell morphology indexes, we showed that the size and shape of the mutants do not display statically significant differences when compared to the wild type ones (Fig. 6b).

## LPS and outer membrane
Similarly, controversy surrounded the cell type of Planctomycetes upon the discovery of their internal membrane organizations[27]. Reports of genes related to diderm bacteria brought arguments to the view that Planctomycetes are derived from diderm bacteria[29]. The functionality of these genes was however unknown. In our data, almost all genes involved in Lipid A synthesis (*lpxABCDKL*) are essential, reinforcing the presence of LPS in *Planctomycetes*[55]. The mechanism to insert beta-barrel proteins in the outer membrane required in diderm bacteria includes the BamABCD complex. BamA is a beta-barrel assembly machinery in the outer membrane, while

BamBCD are accessory proteins. Only *bamA* is essential in our screening, the other genes are either not essential, not found, or have multiple paralogs. Similarly, Lgt and Lnt are involved in the targeting and assembly of lipoproteins. The Sec system involved in secretion is found almost complete in *P. limnophila*, but for the chaperone SecB. The product of *ftsY* is responsible for signal recognition particle receptors, ensuring targeting to the membrane, together with SecYEG. Except for *secG*, all the above-mentioned genes are essential in *P. limnophila* (Sup. Table 1). Altogether, this data adds additional evidence to support the diderm variation of Planctomycetes[29,56].

In conclusion, we report a comprehensive genome-wide study of essential gene content in the model Planctomycetes *P. limnophila*. Our analysis confirms the hidden novel biology of Planctomycetes, emphasizing their interest as emerging model organisms for divergent biology, and revealing the essential genes behind it. We showed that most division and PG synthesis genes usually essential in other bacteria are dispensable in *P. limnophila*. In addition, our data provides further support to the diderm cell type of Planctomycetes. We reveal many essential proteins of unknown function (including DUFs). Those will be the targets of future work to decipher the divergent biology of members of the PVC superphylum. Our study opens the field of genome interrogation for bacteria from the ecologically and biodiverse relevant PVC superphylum.

## Methods
### Bacterial strains and culture conditions
Strains used in this work are listed in Sup. Table 2. *Escherichia coli* strains were grown in Lysogeny broth medium (LB) at 37 °C and *P. limnophila* DSM 3776 in a modified PYGV medium (DSMZ medium 621 [http://www.dsmz.de]: 0.1% yeast extract, 0.1% peptone, 0.1% glucose, 10 mM HEPES (pH 7,5), vitamin solution and Hutners basal salt solution from DSMZ 590 medium). All Planctomycetes were grown at 28 °C. 1% bacto-agar was added for the solid medium. To avoid contamination of the planctomycetes cultures, cycloheximide 50 μg mL$^{-1}$ was added. Cultures were grown aerobically in a shaker (180 rpm). When required, antibiotics were used at the following concentrations (μg mL$^{-1}$): kanamycin (Km) 25 for *E. coli* and 50 for *P. limnophila*, gentamicin (Gm) 10 for *E. coli* and 20 for *P. limnophila* and ampicillin (Ap) 100 for *E. coli* and *P. limnophila*. Growth curve assays were performed in triplicates using only cycloheximide.

## Plasmid description and genetic modification

Plasmids used for gene deletion in a double event of homologous recombination were derived from pEX18Tc vector[57]. To construct knockout plasmids (Sup. Table 3), 700–900 bp upstream and downstream fragments of the target gene were amplified by PCR from genomic DNA using the primer pairs listed in Sup. Table 4. The upstream and downstream fragments digested with the appropriated enzymes were then cloned into pEX18Tc by three-way ligation. Finally, the kanamycin/gentamicin resistance gene amplified from the pUT-miniTn5km/ plasmid[58] was subsequently cloned as a *Bam*HI fragment between the two flanking regions.

Genetic transformation of *P. limnophila* was performed by electroporation. Fresh electrocompetent cells were prepared from 400 mL of a *P. limnophila* culture at $OD_{600}$ of 0.4 in modified PYGV[59]. The cells were washed twice with 100 and 50 mL of ice-cold double-distilled sterile water and once with 2 mL of ice-cold 10% glycerol. Then, the pellet was resuspended in 400 mL of ice-cold 10% glycerol and aliquots of 100 μL were dispensed into 0.1-mm gapped electroporation cuvettes along with 0,5–1 μg of purified plasmid and 1 μL of Type-One restriction inhibitor (Epicenter). The cells were then plated onto modified PYGV plates supplemented with the appropriate antibiotic and were incubated at 28 °C until colony formation after 7–9 days. Colonies were segregated onto fresh selection plates and genotyped by PCR and sequencing.

## Imaging

Bacteria from 2 mL of exponentially growing culture ($OD_{600}$ ~0.4) were harvested (12,000 × *g*, 3 min) and resuspended in 100 μL of fresh medium. A sample of 2 μL was spotted on a glass-bottom dish (MatTek) and covered with a 1% agarose in PBS medium cushion. Brightfield images were acquired using a 100x/1.46 objective through a 1.6x amplification lens and an EMCCD Andor iXon camera mounted on a Zeiss microscope, resulting in a pixel size of 0.1 × 0.1 μm.

## Image analysis

An image analysis workflow was designed for all images using FIJI until a satisfactory segmentation of the cells was achieved. Afterward, the area of the cells was measured and fitted to an ellipse. In order to extract the cell's width and length, we identified them as the ellipse's minor and major axis, calculating subsequently the circularity/roundness. We exported the values of every single cell for further statistical analysis with R/Python.

## Transposase purification

The transposase was purified by the Proteomic facility at the CABD (Seville, Spain) according to Picelli et al.[60] with some modifications. In brief, half a liter of a culture of *E. coli* ER2566 bearing pTXB1-Tn5 plasmid was grown in LB Ap at 37 °C to $OD_{600}$ ~0.6. The culture was then chilled for 15 min to 4 °C, and IPTG was added to 0.5 mM. After continued growth for an additional 3 h at 20 °C, the culture reached $OD_{600}$ 1,4-1,6. Cells were collected by centrifugation at 6000 x *g* 4 °C, and the cell pellet was frozen at −80 °C. The cell pellet was resuspended in 10 mL HEGX (20 mM HEPES-KOH at pH 7.2, 0.8 M NaCl, 1 mM EDTA, 10% glycerol, 0.2% Triton X-100) with complete protease inhibitors and subsequently lysed by sonication. Sonication was carried out for 40 cycles of 5 s on and 5 s off at 40% on a Branson sonicator with a 10-mm tip on ice.

The lysate was pelleted at 8000 x *g* for 20 min at 4 °C and 0.53 mL 10% neutralized PEI was added to the supernatant dropwise on a magnetic stirrer, and the precipitate was removed by centrifugation at 10,000 x g for 20 min at 4 °C. The supernatant was loaded on a 2,5 mL chitin column prewashed with 20 mL HEGX at 0.4 mL min⁻¹ in HEGX. The column was washed at 1 mL min⁻¹ with 50 mL HEGX, following which another 20 mL HEGX, 100 mM DTT was added to the top of the column bed. The column was left closed for 48 h at 4 °C to affect

cleavage of Tn5 from the intein. Elution was done in 1,5 mL aliquots and the protein concentration was tested using a Bradford assay (Bio-Rad protein assay).

The most concentrated fractions were pooled and dialyzed versus two changes of one liter of 2X Tn5 dialysis buffer (100 mM HEPES-KOH at pH 7.2, 0.2 M NaCl, 0.2 mM EDTA, 2 mM DTT, 0.2% Triton X-100, 20% glycerol) and freeze at −80 °C adding glycerol to a final concentration of 60%.

## Transposome assembly

Transposome assembly was done following the protocol described by Goryshin et al.[35], with some modifications. Transposomes were formed by incubating Km transposon DNA (up to 50 μg mL⁻¹) with 10 μg mL⁻¹ Tn5 transposase for 1 h at 37 °C in a 20 μL reaction volume in transposon buffer (27.5 mM Tris-HCl, pH 7.5, 50 mM NaCl, 0.075 mM EDTA, 0.5 mM dithiothreitol, 0.05% Triton X-100, and 50% glycerol). Transposomes were stored at −20 °C until used. Km gene was amplified using primers Km IS fwd and Km IS rv (Sup. Table 4).

## Transposon library construction

Wild-type *Planctopirus limnophila* DSM3776 was used for the construction of a transposon library. Fresh electrocompetent cells were prepared as described above. Aliquots of 100 μL of competent cells were dispensed into 0.1-mm gapped electroporation cuvettes along with 1 μL of transposome and 1 μL of Type-One restriction inhibitor (Epicenter). Electroporation was performed with a Bio-Rad Micropulser (Ec3 pulse, voltage [V] 3.0 kV). Electroporated cells were immediately recovered in 1 mL of cold-modified PYGV and incubated at 28 °C for 2 h with shaking. Transposon mutants were selected by growth onto modified PYGV supplemented (100 mm plates) with cycloheximide 50 μg mL⁻¹ and kanamycin 50 μg mL⁻¹ at 28 °C growing for 10 days. Plates were swabbed and approximately 1.1 million colonies were pooled together and stored at 80 °C. Clones were not replica plated prior to harvest, thus, clones with limited growth may have also been included. To verify Tn5 insertions and their locations, the DNA of random candidates was isolated using the Wizard Genomic DNA Purification Kit (Promega) and analyzed by semi-random polymerase chain reaction (PCR)[61]. Genomic DNA was used as the template DNA in a 20 μL PCR mixture containing primer Map Tn5 A fwd, and either primer CEKG 2A, CEKG 2B, or CEKG 2 C; 1 μL of a 1:5 dilution of this reaction mixture was used as the template DNA for a second PCR performed with primers Map Tn5 B fwd and CEKG 4. For the first reaction, the thermocycler conditions were 95 °C for 2 min, followed by six cycles of 95 °C for 30 s, 42 °C for 30 s (with the temperature reduced by 1 °C per cycle) and 72 °C for 3 min and then 25 cycles of 95 °C for 30 s, 65 °C for 30 s and 72 °C for 3 min; for the second reaction, the thermocycler conditions were 30 cycles of 95 °C for 30 s, 65 °C for 30 ms and 72 °C for 3 min. The DNA of purified PCR products (GFX PCR DNA and Gel Band Purification Kit GE Healthcare) was sequenced by using primer Map Tn5 B fwd.

## Library preparation and sequencing

DNA was extracted from seven samples of the transposon library to generate TraDIS data. The extractions were done using the Wizard Genomic DNA Purification Kit (Promega). The concentration and quality of the genomic DNA were checked using Picogreen method (Qubit). Library preparation and sequencing were performed by Illumina MiSeq by Fasteris (Switzerland).

## Sequencing data mapping

Raw data were collected and analyzed using a series of custom scripts. In order to map the sequence data to the genome (GenBank codes for genome and plasmid, respectively: CP001744.1 and CP001745.1), reads were filtered for the ones containing the exact sequence of the transposon termini with the program cutadapt cutadapt[62] (v4.4) and the

following parameters: -g XNNNNNGTTCGAAATGAGATGTGTATAA-GAGACAG -e 0 -O 34. The adaptor sequence was trimmed from the reads passing the filter with the program cutadapt and the following parameters: -m 15 -a ATGGAATTCTCGGGTGCCAAGG. The resulting reads were mapped to the genome with bowtie2[63] (v2.5.1). The subsequent steps of conversion from SAM (sequence alignment/map) files to BAM (binary version of SAM) files, and the requisite sorting and indexing, were done using SAMtools[64] (v1.17). Duplicated reads were removed with picard-2 (v3.0.0) (https://broadinstitute.github.io/picard/). The following data treatments were realized with in-house Python scripts. Data were inspected manually using the IGV genome browser[65] (v2.16.1).

### Essential gene prediction

We followed the method proposed by Goodall et al.[10], with minor modifications. Each gene's insertion rate was calculated as the total number of insertions per gene divided by the total gene size. As in the reference, the distribution presented two modes. Each one was adjusted to a different model, the first section was adjusted to the exponential distribution and the second to the gamma distribution using packages MASS (v7.3-60) and fitdistrplus(v1.1-11) in R (v3.10.12). Unlike the original method, the distribution was separated into three sections due to the high overlapping in the transition between modes. The insertion index cut-off values between these sections were manually established in 0.013, 0.022, and 0.25, based on the observation of data. The likelihood that a gene belonged to each of these two distributions was calculated. The ratio of both likelihoods was used to calculate a log-likelihood score.

Genes were assigned as essential by TraDIS if the log of the likelihood ratio is higher than $\log_2(12)$, as non-essential if this is less than $\log_2(-12)$, and unclear if this is between these values.

A Python (v3.10.12) script was used to interrogate the TraDIS unclear genes. A window of 300 bp started the insertion count at the beginning of the gene and slid every 150 bp. A window was labeled as essential when no inserts were found inside the window.

### Distribution of *P. limnophila* genes by functional class

The annotation of COG functional label from *P. limnophila* was retrieved from proGenomes[66] (v2.1) [http://progenomes2.embl.de/]. This annotation was plotted in different bar charts using the packages ggplot2 (v3.4.2) in R (v3.10.12). The COG functional labels from genes with more than one label were separated and taken into account as different to count the number of total labels.

### *P. limnophila* and *E. coli* orthologs

A reciprocal protein BLAST[67] (BLASTP version 2.12.0+) was carried out between the *P. limnophila* (accession no. CP001744.1 and CP001745.1) and *E. coli* K-12 BW25113 (accession no. CP009273.1) proteomes. The results were filtered using a cut-off E-value of $1 \times 10^{-5}$ and a query and subject coverage of at least 70%. We then applied additional filter criteria to remove proteins with paralogs. The ortholog proteins found were classified based on the comparative essentiality reported.

### *P. limnophila* conservation

A protein BLAST was carried out for each *P. limnophila* protein (accession no. CP001744.1 and CP001745.1) against a custom BLAST protein database created from the KEGG database [https://www.genome.jp/kegg/] (v2105)[68–70] using only prokaryotic organisms (Sup. Data 8).

Subsequently, subject coverage was calculated for each of the results and those with a query/subject coverage greater than or equal to 70% and with an E-value less than or equal to $1 \times 10^{-5}$ were selected. Hits were grouped according to species to minimize differences at the strain level.

### Phyletic distribution

The orthology groups of the genes of *P. limnophila* and *E. coli* were retrieved from EggNOG Database 5[45] [http://eggnog5.embl.de/#/app/home]. For each of these ortholog groups, the frequency of occurrence *per* class was calculated for over 5, 454 organisms collected in the KEGG database [https://www.genome.jp/kegg/]. Classes with less than five organisms were not considered, reducing the number to 5277 organisms in a total of 50 classes (Sup. Data 9).

For those groups that appeared more than once in the same organism, only one occurrence was considered to denote their absence or presence. For the assignment of ortholog groups to these selected organisms, EggNOG-mapper[45,46] (v2.1.12) was used with default values.

With the frequency data of orthologous groups in the different phylogenetic classes, a heatmap was made using package Pheatmap (v1.0.12) (Kolde, Pheatmap: Pretty Heatmaps. R package version 1.0.12 (2019). A hierarchical clustering was made with the complete linkage method and Euclidean distance by this library. The number of clusters was designated manually based on tree observations.

### Cell division-related proteins in *P. limnophila*

Hidden Markov Models (HMM) of all division-related genes were created using HMMER (v3.3.2) *hmmbuild* function from the multi-sequence alignment of the protein reported by Pierre S. Garcia et al.[71]. Then, *hmmsearch* was used to find orthologs of division proteins in *P. limnophila*, using the model mentioned above. These models were run on the models themselves and the mean and standard deviation were calculated. These results were filtered by the bit-score values, considering the results with a bit-score value higher than 50% of the mean minus one standard deviation, followed by manual inspections.

### Paralogous detection

All proteins from *P. limnophila* were compared against each other using BLAST searches and subsequently grouped into paralogous groups considering the following criteria.

1. Relative sequence lengths of the proteins, where we request that the size of the smaller protein must be at least 60% of the larger one.
2. Relative bit-scores of the BLAST comparisons, where given two proteins, A and B, the bit-score values of each protein compared to themselves were evaluated. The largest value of these comparisons, named Max_bit_score, was used as a reference value. For two proteins A and B that belong to the same COG group, the relationship between the resulting bit-score of comparing these proteins must be at least 0.1 of the Max_bit_score value. For cases where A and B do not share the same COG group, this cut-off value was set to 0.2 of the Max_bit_score value.
3. Similarity relationships between potential members of a group of paralogous proteins.

Given a protein p1, within the set of proteins p2, p3,..., pN, of a proteome, all those proteins that meet the two previous criteria are identified. This first set of proteins was considered as the first-order similarity-neighbors of the p1 protein. Subsequently, for each first-order similarity neighbor of the p1 protein, their corresponding first-order similarity-neighbors were identified. The non-redundant set of these proteins was called the second-order similarity-neighbors of the p1 protein. In the first instance, the protein with the highest number of second-order similarity-neighbors was considered the proteome's first protein reference. Each one of its second neighbors is considered to be part of the paralog group of the p1 reference protein if the said protein has several second-order neighbors equal to or greater than 20% of the number of second-order neighbors that the p1 reference protein has. Since proteins in a proteome can only belong to one paralog group, once a protein is assigned to a group, it is no longer considered in subsequent clustering analyses and is therefore removed from the

protein list. The process described above is repeated cyclically until all the proteins in the proteome are assigned to a paralog group.

## Image analysis

The image analysis workflow runs as follows under FIJI software (v2.9.0): Image Acquisition → Subtract Background → Gaussian Blur → Invert → Enhance Contrast → Unsharp Mask → SFC H Watershed Thresholding using seeds → Convert to mask → Binary Watershed → Analyze Particles. We set a filter of size using the upper and lower limits of 0.3 to 2 $\mu m^2$, since we consider that any particle out of this limit was not a cell. The presence of this filter does not affect the distribution of the results as visible in the plots (Fig. 6b).

## Reporting summary

Further information on research design is available in the Nature Portfolio Reporting Summary linked to this article.

## Data availability

The raw sequencing data from the TraDIS library generated in this study have been deposited in the Figshare repository (https://doi.org/10.6084/m9.figshare.24249346). All other data generated in this study are provided in the Supplementary Information and Source Data files. Source data are provided in this paper.

## Code availability

The custom scripts used for analyses in this study are available in the repository https://github.com/dmoypal/TraDIS_in_P.limnophila.

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

## Acknowledgements

Thanks to Ildefonso Cases (CABD computational unit) for bioinformatic services and Alejandro Campoy (CABD Advanced Light Microscopy and Imaging Facility). E.R.M. is supported by Fondo Europeo de Desarrollo Regional (FEDER, Confunding percentaje FEDER 80%) and Consejería de Economía, Conocimiento, Empresas y Universidad (Junta de Andalucía) (UPO1381049); D.M.P. by Junta de Andalucía Predoctoral Grant (Predoc_00913); V.H. by Fundação para a Ciência e a Tecnologia, Portugal (2022.11400.BD) and D.P.D. by Mineco (Grant No. PID2020-119733GB-I00) and the "Moore-Simons Project on the Origin of the Eukaryotic Cell" (Grant No. 9733/ https://doi.org/10.37807/GBMF9733).

## Author contributions

E.R.M. and D.P.D. designed the study. D.M.P., E.M., and D.P.D. performed in silico analysis. E.R.M. and V.H. realized the molecular biology, the transposon mutant library, constructed the deletion mutants, and performed the physiological assays. All authors analyzed and interpreted data and contributed to the writing of the manuscript.

## Competing interests

The authors declare no competing interests.
