## [Peer Review File · Nature Communications]

Reviewers' Comments:

Reviewer #1:

Remarks to the Author:

Summary

Rivas-Marin et al. take a functional genomics approach to identify and characterize essential genes in the model Planctomycetes species, *Planctopirus limnophila*. They first experimentally determined essential genes in *P. limnophila* using TraDIS, finding a very large number of essential genes (783 or 18% of the genome). They then compared the *P. limnophila* essentials to gold-standard essentials from *Escherichia coli*, which highlighted a lack of essentiality for peptidoglycan (PG) synthesis genes in *P. limnophila*. The authors validated the non-essentiality of PG genes by constructing deletion strains; the majority of which showed no detectable growth phenotype. Overall, the manuscript provides a valuable resource for the Planctomycetes field but requires important revisions/additions.

General Comments

1. The number of essential genes in *P. limnophila* seems abnormally high compared to other bacteria. TraDIS/Tn-seq tends to overestimate the number of essentials compared to gold-standard deletion analysis, as was shown in the Goodall reference and further investigated in Choe et al. (10.1128/msystems.00896-22). Even considering that, as the authors note, the number/fraction of *P. limnophila* essentials is much higher than for well-studied bacteria (12.19 % for *Caulobacter crescentus* and 8.3 % for *E. coli*). The authors suggest that there are biological reasons for the high number of essentials (e.g., growth medium), but I don't find that explanation completely satisfying given that there are hundreds more than I typically expect to see. One issue might be in the way the authors quantified whether genes were essential, which has sophistication, but is also somewhat arbitrary. I suggest that the authors provide a false discovery rate for essential genes at various cutoffs so that readers can make their own interpretation of how many potential false positives are included at a given cutoff.
2. The TraDIS experimental follow-ups are limited and not entirely convincing. The non-essentiality of conserved PG synthesis/cell division genes in *P. limnophila* is intriguing but requires further characterization. First, because the PG/div deletion strains show little or no apparent growth phenotypes, the authors need to provide additional evidence they are true deletion strains--the methods state that the authors confirmed the mutants by PCR/sequencing, which should be shown in the supplement. Second, the mutants need at least some additional characterization. For instance, is cell shape affected by the loss of PG/div genes as has been shown for other bacteria? Simple light microscopy could provide important insight into the roles of these genes that seem to be retained by *P. limnophila*, despite the fact that they are not essential.
3. Genetics alone cannot be used to determine if a species is a diderm (Gram-negative). Whether or not *P. limnophila* is a diderm is a question best solved with imaging techniques, rather than genetics. For instance, *Acinetobacter baumannii* is a diderm that does not require LPS/LOS synthesis genes for viability, so one cannot conclude this physical feature of the cell through genetics alone. The authors should back off of this claim, make it clear that their claim is speculative, or provide additional evidence supporting the claim.

Specific comments

1. How many replicates were performed for the TraDIS screen and the growth curves?
2. The statement, "More broadly, our data reveal that the mere presence of the gene is not enough to infer a canonical functional pathway," is difficult to parse, especially because the meaning of "canonical functional pathway" is unclear. Are the authors simply trying to say that just because a gene is essential in one bacterium doesn't mean it's essential in another, or are they trying to make a more complex point about gene functions not being completely identical?

3. Orthology analysis between *Planctopirus limnophila* and *Escherichia coli* shows few genes in common. If this is used as evidence to validate the TraDIS screen, the concept needs further elaboration.

4. Figures need more detailed explanations – Legends should provide clarification of their supporting points. Axes are generally hard to interpret, and *Planctopirus limnophila* is not easily found in phylogenetic plots.

5. Figure 1: The reported 1,158,840 reads seem to indicate insufficient coverage for approximately 400,000 insertion sites. However, the diagrams suggest high levels of coverage.

6. Figure 2: The paper should specifically reference the Goodall paper (and possibly others) related to the concept of domain essentiality.

7. Figure 3: How do the observed functional trends differ among various bacteria; i.e., would one expect the same distribution of essential COG functions in *E. coli*?

8. Figure 4: The screenshot is low quality and difficult to read. Consider changing locus tags to gene names for annotated genes. Clarify what the reader should infer from this figure.

9. Figure 5: *Planctopirus limnophila* is not easily found in phylogenetic plots. Some heatmap sections behave differently from the rest, drawing the reader's attention, but the interpretation of these spots is unclear. The x-axis is difficult to read in both A and B.

10. Figure 6: Please provide an interpretation for the growth curve. Is the reader supposed to conclude that all strains grow similarly to wild type, or do the *rodA/mraW2* strains have a growth defect?

Reviewer #2:

Remarks to the Author:

Summary:

In this study, Rivas-Marin et al. present results showing the essentiality of the *P. limnophila* genome in vitro. Using transposon mutagenesis transposon directed insertion site sequencing (TraDIS, also known as Transposon Sequencing, or TnSeq) they create a large library of *P. limnophila* mutants, harboring random transposon insertions, disrupting the genetic function of that location. After sequencing and determining the regions able to withstand disruption, they define the essential and non-essential genes in this organism, and compare their results with homologous genes in other organisms.

The high saturation of their mutant library and it being (as far as I can tell) the first report of genome-wide essentiality in this organism, makes this an important and relevant contribution to the literature. However, there are some comments I would like to see addressed before recommending the manuscript be published:

Major Comments:

- Authors say they "introduced the concept of domain essentiality" to describe certain cases of genes. This should be rephrased to avoid confusion, since this is not a new concept. The concept of domain essentiality has been introduced to the literature long ago..

- Authors classify genes in three different categories ("essential", "non-essential" and "unclear"), based on cut-off values. They say these cutoff were determined after visual inspection. Figure S1 should have these cutoffs clearly marked to let the reader easily inspect them as well. Did the authors attempt a more rigorous way of determining these cutoffs or splitting these values as a mixture of 3 distributions instead of 2?

- After classifying genes into these 3 categories, the authors then delve deeper and classify genes as "domain-essentials" (genes containing long stretches without a transposon insertion). However, they keep the original classification for these genes i.e. a gene called "unclear" is also called "domain essential". To me, this is confusing. If a gene is a domain-essential, then it's no longer unclear. Similarly, if a gene has an essential domain, then is the function of (part of) the protein not essential?

Although not necessary, I think it would help readers understand the manuscript better if domain-essentiality were a refining of the original calls and either all treated as a subset of essential genes (i.e. mark all domain-essential genes as now being essential) or if they were kept as a 4th category of essentiality (e.g. "essential", "non-essential" and "unclear" and "domain-essential").

Minor Comments:

- There is a "The" missing at the beginning of the sentence introducing TraDIS, i.e. "Transposon-directed insertion site sequencing (TraDIS) method".

- The last few sentences in the penultimate paragraph of the Introduction i.e. "These and other features have recently raised interest which translates into the exponential ..." should be re-written as they are confusingly phrased.

- The first sentence of the last paragraph in the "Comparative essentiality" section ("In addition, to reflect differences...") seems to be grammatically incorrect.

Reviewer #3:

Remarks to the Author:

In this manuscript, the authors report on their findings subsequent to the completion of a rather extensive transposon library in *Planctopirus limnophila*. This work follows closely their previous study in which they demonstrated that a number of genes (*ftsI*, *ftsW* and *mreB*) associated with cell division and localized peptidoglycan biosynthesis are not essential for the growth of *P. limnophila*. Here they expand of this initial finding and demonstrate that a number of genes from the division and cell wall (*dcw*) cluster also do not appear to be essential in *P. limnophila*. This is in stark contrast to genes associated with LPS / Lipid A biosynthesis, which all appear to be essential for the organism.

The major strengths of the manuscript are:

1) For the most part the methodology utilized is sound, and the genetic analysis was conducted appropriately

2) The manuscript will prove impactful in its future use in studying the contributions of 'essential' and 'non-essential' genes to the development and survival of *P. limnophila* and potentially aiding in answering lingering questions on it means of cell division in the absence of many elements of more canonical bacterial systems (eg. *FtsZ*).

3) The authors did a good job of confirming transposon mutants with complete deletional strains

The major weaknesses of the manuscript are:

1) The authors' use of the term 'essential' (as in essential genes) needs some context. Essentiality is conditional: what is essential on the surface of a fresh water lake in Germany is very different from what is essential in modified PYGV medium on a shaker. This fact has direct implications for many of the transposon mutants identified in the organism's transporters / metabolic pathways, and it may also impact the interpretation of the peptidoglycan biosynthesis mutants as well.

2) The manuscript on the whole was rather heavy on bioinformatics and light on experimentation. With all of these wonderful new mutants available, it seems a shame that the only experiment conducted was the demonstration of growth / viability of a couple of strains in a gene set already suspected of not being essential for laboratory growth. Bigger questions could have been easily addressed, such as if these genes are not essential for growth / replication under optimal growth conditions, what about under non-optimal (and likely more physiologically relevant) growth / survival conditions?

3) the use of optical density rather than a more accurate measure for bacterial growth (cfu counts) is a notable weakness; many things affect OD600 values that have nothing to do with cell division and/or bacterial numbers

Minor comments:

- please add line numbers for subsequent manuscript drafts as it aide with review

- please add additional detail as far as the methods by which the transposon library was prepared. The authors need to clarify that colonies were individually picked during construction, passaged separately, and then then pooled later or whether plates containing colonies were simply swabbed and pooled together at the time of harvesting. This is important, as the existence of a large number of metabolic mutants could be explained by their uptake of primary and secondary metabolites from other living / dead bacteria on the initial agar plate

- the authors bring up the potential for compensatory effects of paralogs encoded in the genome; this could be easily demonstrated by assessing the loss of enzymatic / transport activity in the transposon mutants in question, perhaps a few choice examples

"More broadly, our data reveal that the mere presence of the gene is not enough to infer a canonical functional pathway."

- this is an overly broad interpretation of the data presented. The authors have not demonstrated that any canonical pathway is not functional. They merely observe that some do not appear to be essential based off of their transposon saturation library. They also do not demonstrate (either experimentally or citing the primary literature) functionality of any of the original gene products (something that has proven critical for the exploration of similar pathways in other bacterial systems, such as the Chlamydiales).

- The "Homology with Chlamydiae" section does not appear to add much to the authors' story. It is well known that translation, ribosomes, and biogenesis genes are essential in *C. trachomatis*, so it is unclear what 'revealing potential targets to control this important human pathogen' the authors are referring to. Additionally, all PG biosynthesis genes examined to date (including MreB, FtsI, and PBP2) are essential for *C. trachomatis* growth in HeLa / HeP2 cells. One of the more interesting similarities between the two organisms is that both appear to divide via a mechanism akin to budding, but while the process appears to be asymmetric in *P. limnophila*, it is actually a symmetric process in *Chlamydia* tied directly to septal peptidoglycan formation (Liechti 2016, Abdelrahman 2016). This is likely a key

defining feature between these two bacterial groups that the authors may wish to highlight. If peptidoglycan biosynthesis is not important for septal formation in *P. limnophila*, what else might be providing the selection to maintain so many PG / 'cell division' genes in the genome?

Dear Dr Rivas-Marin,

Thank you again for submitting your manuscript "Essential gene complement of the non-model bacterium *Planctomycetes Planctopirus limnophila* reveals the genetic components of divergent biology" to Nature Communications. We have now received reports from 3 reviewers and, after careful consideration, we have decided to invite a major revision of the manuscript.

As you will see from the reports copied below, the reviewers raise important concerns. We find that these concerns limit the strength and potential impact of the study, and therefore we ask you to address them with additional work. Without substantial revisions, we will be unlikely to send the paper back to review.

In particular:

- Reviewer #1 highlights the unusually high number of genes classified as essential in your work and requests statistical analysis including false discovery rates. Reviewer #2 also makes comments about your choice of essentiality cut-offs.

We have worked on those two points. Concerning the first, we have sequenced more data, more than doubling the number of reads, reaching 2,863,686 clean reads (from 8,132,446 raw) and we also have commented on the condensation of the DNA possibly hindering transposition, as suggested by the reviewer. Concerning the second point, we have provided various lists of essential genes by considering different cut-offs (Dataset 3), as suggested by reviewer #1. You will see that the changes in numbers of essential genes are minor and do not significantly alter our results, while providing more support.

- All three referees notice several overstatements and overstretched conclusions that should be removed or toned down.

In agreement, we have tuned down the manuscript.

- Reviewer #1 and #3 recommend additional phenotypic characterization of deletion strains, and we agree that this addition would strengthen the case for publication in our journal.

We have now performed light microscopy assays and growth curves of all our mutants in order to include morphological characterization of them.

REVIEWER COMMENTS

Reviewer #1 (Remarks to the Author):

Summary

Rivas-Marin et al. take a functional genomics approach to identify and characterize essential genes in the model *Planctomycetes* species, *Planctopirus limnophila*. They first experimentally determined essential genes in *P. limnophila* using TraDIS, finding a very large number of essential genes (783 or 18% of the genome). They then compared the *P. limnophila* essentials to gold-standard essentials from *Escherichia coli*, which highlighted a lack of essentiality for peptidoglycan (PG) synthesis genes in *P. limnophila*. The authors validated the non-essentiality of PG genes by constructing deletion strains; the majority of which showed no detectable growth phenotype. Overall, the manuscript provides a valuable resource for the *Planctomycetes* field but requires important revisions/additions.

We thank the reviewer for this nice summary of our work. We would like to highlight that our focal point is mainly the non-essential genes. Essentials are very interesting but subject to artifacts, technical, biological and statistical. Non-essential are more robust and the ones we focus on are mostly genes involved in peptidoglycan synthesis and cell division.

General Comments

1. The number of essential genes in *P. limnophila* seems abnormally high compared to other bacteria. TraDIS/Tn-seq tends to overestimate the number of essentials compared to gold-standard deletion analysis, as was shown in the Goodall reference and further investigated in Choe et al. (10.1128/msystems.00896-22). Even considering that, as the authors note, the number/fraction of *P. limnophila* essentials is much higher than for well-studied bacteria (12.19 % for *Caulobacter crescentus* and 8.3 % for *E. coli*). The authors suggest that there are biological reasons for the high number of essentials (e.g., growth medium), but I don't find that explanation completely satisfying given that there are hundreds more than I typically expect to see. One issue might be in the way the authors quantified whether genes were essential, which has sophistication, but is also somewhat arbitrary. I suggest that the authors provide a false discovery rate for essential genes at various cutoffs so that readers can make their own interpretation of how many potential false positives are included at a given cutoff.

We believe that we provide a fair comparison with other organisms and methods. Our statistical method is exactly the same as the one of Goodall *et al.*, providing a methodological copy of their work, as exact as possible. However, we agree that some parameters might seem arbitrary. As suggested, we now provide various lists of essential genes calculated with different cut-offs (Dataset S3). The numbers are not significantly different, providing support to our results.

As can be appreciated, the majority of the genes affected by the changes are the unclear ones, and all those in the range of overlap between distributions (Dataset S3). In addition, the number of genes affected is low, 110.

We also appreciate the suggestion to consider the reference Choe *et al.*, 23. In particular, the possibility that some of the genomic regions might be protected by nucleotide binding proteins, artefactually increasing the number of genes deemed as essentials. Given the high degree of genome condensation in *Planctomycetes*, this is a very likely possibility. We have now commented on this possibility in the text.

2. The TraDIS experimental follow-ups are limited and not entirely convincing. The non-essentiality of conserved PG synthesis/cell division genes in *P. limnophila* is intriguing but requires further characterization. First, because the PG/div deletion strains show little or no apparent growth phenotypes, the authors need to provide additional evidence they are true deletion strains--the methods state that the authors confirmed the mutants by PCR/sequencing, which should be shown in the supplement. Second, the mutants need at least some additional characterization. For instance, is cell shape affected by the loss of PG/div genes as has been shown for other bacteria? Simple light microscopy could provide important insight into the roles of these genes that seem to be retained by *P. limnophila*, despite the fact that they are not essential.

We now included all PCR assays performed to verify the deletions in the supplementary material (Fig. S5). This, together with the TraDIS data, should be enough to provide confidence in the mutants.

In addition, we have now performed light microscopy, where we analysed some morphological parameters of the mutants (Fig. 6B). No significant changes were detected.

3. Genetics alone cannot be used to determine if a species is a diderm (Gram-negative). Whether or not *P. limnophila* is a diderm is a question best solved with imaging techniques, rather than genetics. For instance, *Acinetobacter baumannii* is a diderm that does not require LPS/LOS synthesis genes for viability, so one cannot conclude this physical feature of the cell through genetics alone. The authors should back off of this claim, make it clear that their claim is speculative, or provide additional evidence supporting the claim.

Agreed, we have now tuned down this part as suggested, in agreement with a more general down-tuning of the manuscript.

Specific comments

1. How many replicates were performed for the TraDIS screen and the growth curves?

For TraDIS assay, one transposon library was realized and seven DNA samples for independent sequencing libraries were submitted. Growth curves were performed in triplicates. This information has been included in the Material and Methods section.

2. The statement, "More broadly, our data reveal that the mere presence of the gene is not enough to infer a canonical functional pathway," is difficult to parse, especially because the meaning of "canonical functional pathway" is unclear. Are the authors simply trying to say that just because a gene is essential in one bacterium doesn't mean it's essential in another, or are they trying to make a more complex point about gene functions not being completely identical?

See also the comment of reviewer #3 on the same point. We would say that both are correct. Essentiality in one bacterium does not imply essentiality in another, but also function is not always identical between ortholog genes, although likely to be related. We have clarified that part in the manuscript.

3. Orthology analysis between *Planctopirus limnophila* and *Escherichia coli* shows few genes in common. If this is used as evidence to validate the TraDIS screen, the concept needs further elaboration.

This experiment was not done to validate the TraDIS experiment but to compare two evolutionarily distant species. This distance is reflected only in the low number of orthologues found. However, please note that this is only the number of direct, one-to-one orthologues. In fact, there are 1,045 *P. limnophila* genes with orthologues in *E. coli* with the criteria used. However, most of them show more than one orthologue, making a direct comparison difficult.

4. Figures need more detailed explanations – Legends should provide clarification of their supporting points. Axes are generally hard to interpret, and *Planctopirus limnophila* is not easily found in phylogenetic plots.

Modified. Please note that the high resolution is now provided and that *P. limnophila* has been highlighted in the phylogenetic plot (Fig. 5).

5. Figure 1: The reported 1,158,840 reads seem to indicate insufficient coverage for approximately 400,000 insertion sites. However, the diagrams suggest high levels of coverage.

We believe this number of reads is sufficient for the proposed analysis (please notice that reviewer #2 agrees). They are in the range reported by other publications. In fact, Choe *et al.*, report only twice that, in the same order of magnitude for the same number of insertion sites.

Goodall *et al.* have 4-5 times this, same order of magnitude, for slightly less unique insertion sites. We have anyway resequenced and now provide 2,863,686 reads. Additive curves also show that we are reaching saturation.

6. Figure 2: The paper should specifically reference the Goodall paper (and possibly others) related to the concept of domain essentiality.

We have added the reference, thanks.

7. Figure 3: How do the observed functional trends differ among various bacteria; i.e., would one expect the same distribution of essential COG functions in *E. coli*?

Figure 3 illustrates a basic descriptive analysis where we aim to show an overview of the essentiality grouped by functional categories in *P. limnophila*. We did the same analyses with *E. coli* to compare them (see figure below). However, we have not added it to the manuscript because we already have a detailed comparison between them in the section "Comparative essentiality", it would be redundant. In this section, we have also considered redundancies between orthologs.

The graph shows that both distributions are similar, with some slight differences related to the specific biology of each organism, highlighting the difference in the functional category "Energy production and conversion" and "Cell wall/membrane/envelope biogenesis". However, these examples are mentioned in the section "Comparative essentiality".

8. Figure 4: The screenshot is low quality and difficult to read. Consider changing locus tags to gene names for annotated genes. Clarify what the reader should infer from this figure.

We have added the locus tags to the gene names for annotated genes to improve readability. As mentioned before, we now provide high-resolution figures in general.

Regarding the interpretation of the figure, we have clarified it.

9. Figure 5: Planctopirus limnophila is not easily found in phylogenetic plots. Some heatmap sections behave differently from the rest, drawing the reader's attention, but the interpretation of these spots is unclear. The x-axis is difficult to read in both A and B.

We have improved the readability of the X-axis in Figure 5. We have also highlighted *P. limnophila* in the phylogenetic plots for easier identification.

Regarding the heatmap's interpretation, this plot aims to present a graphical overview of the conservation of the essential genes. This broad overview is not pretending to pinpoint small details. Readers should be looking at the big picture.

10. Figure 6: Please provide an interpretation for the growth curve. Is the reader supposed to conclude that all strains grow similarly to wild type, or do the rodA/mraW2 strains have a growth defect?

Thanks for the comment. We have provided an interpretation of the growth curve in the manuscript.

Reviewer #2 (Remarks to the Author):

Summary:

In this study, Rivas-Marin et al. present results showing the essentiality of the *P. limnophila* genome in vitro. Using transposon mutagenesis transposon directed insertion site sequencing (TraDIS, also known as Transposon Sequencing, or TnSeq) they create a large library of *P. limnophila* mutants, harboring random transposon insertions, disrupting the genetic function of that location. After sequencing and determining the regions able to withstand disruption, they define the essential and non-essential genes in this organism, and compare their results with homologous genes in other organisms.

The high saturation of their mutant library and it being (as far as I can tell) the first report of genome-wide essentiality in this organism, makes this an important and relevant contribution to the literature. However, there are some comments I would like to see addressed before recommending the manuscript be published:

Major Comments:

- Authors say they "introduced the concept of domain essentiality" to describe certain cases of genes. This should be rephrased to avoid confusion, since this is not a new concept. The concept of domain essentiality has been introduced to the literature long ago.

In agreement with similar comments of the reviewer #1, we have modified this statement.

- Authors classify genes in three different categories ("essential", "non-essential" and "unclear"), based on cut-off values. They say these cutoff were determined after visual inspection. Figure S1 should have these cutoffs clearly marked to let the reader easily inspect them as well. Did the authors attempt a more rigorous way of determining these cutoffs or splitting these values as a mixture of 3 distributions instead of 2?

We did not attempt to fit a mixture of 3 distributions in order not to modify the original method (Goodall *et al.*). See also the comment of reviewer 1, for which we provide a false discovery rate at various cut-offs (Dataset S3). We have also added the cut-offs to the Fig. S1.

- After classifying genes into these 3 categories, the authors then delve deeper and classify genes as "domain-essentials" (genes containing long stretches without a transposon insertion). However, they keep the original classification for these genes i.e. a gene called "unclear" is also called "domain essential". To me, this is confusing. If a gene is a domain-essential, then it's no longer unclear. Similarly, if a gene has an essential domain, then is the function of (part of) the protein not essential?

We agree with the reviewer. We now provide a merged definition in the text. Please see also our answer to the next comment. We consider that when a gene has an essential domain, the protein is essential.

Although not necessary, I think it would help readers understand the manuscript better if domain-essentiality were a refining of the original calls and either all treated as a subset of essential genes (i.e. mark all domain-essential genes as now being essential) or if they were kept as a 4th category of essentiality (e.g. "essential", "non-essential" and "unclear" and "domain-essential").

We agree with the reviewer, now we provide a merged definition in the text. We also show in Dataset S2 the ones that have changed from "non-essential" or "unclear" to "domain-essential" and, thus, "essential" genes. Additionally, we have included a specific dataset (Dataset S4) with all the "domain-essential" genes with its TraDIS label and its function that complement the Supplementary Figure S2.

Minor Comments:

- **There is a "The" missing at the beginning of the sentence introducing TraDIS, i.e. "Transposon-directed insertion site sequencing (TraDIS) method".**

Edited. Thanks.

- **The last few sentences in the penultimate paragraph of the Introduction i.e. "These and other features have recently raised interest which translates into the exponential..." should be re-written as they are confusingly phrased.**

We have clarified the statement. Thanks.

- **The first sentence of the last paragraph in the "Comparative essentiality" section ("In addition, to reflect differences...") seems to be grammatically incorrect.**

We thank the reviewer for the comment, the sentence was misleading. The sentence has been corrected.

Reviewer #3 (Remarks to the Author):

In this manuscript, the authors report on their findings subsequent to the completion of a rather extensive transposon library in *Planctopirus limnophila*. This work follows closely their previous study in which they demonstrated that a number of genes (*ftsI*, *ftsW* and *mreB*) associated with cell division and localized peptidoglycan biosynthesis are not essential for the growth of *P. limnophila*. Here they expand on this initial finding and demonstrate that a number of genes from the division and cell wall (*dcw*) cluster also do not appear to be essential in *P. limnophila*. This is in stark contrast to genes associated with LPS / Lipid A biosynthesis, which all appear to be essential for the organism.

The major strengths of the manuscript are:

- 1) For the most part the methodology utilized is sound, and the genetic analysis was conducted appropriately
- 2) The manuscript will prove impactful in its future use in studying the contributions of 'essential' and 'non-essential' genes to the development and survival of *P. limnophila* and

potentially aiding in answering lingering questions on it means of cell division in the absence of many elements of more canonical bacterial systems (eg. FtsZ).

3) The authors did a good job of confirming transposon mutants with complete deletional strains

The major weaknesses of the manuscript are:

1) The authors' use of the term 'essential' (as in essential genes) needs some context. Essentiality is conditional: what is essential on the surface of a fresh water lake in Germany is very different from what is essential in modified PYGV medium on a shaker. This fact has direct implications for many of the transposon mutants identified in the organism's transporters / metabolic pathways, and it may also impact the interpretation of the peptidoglycan biosynthesis mutants as well.

We agree with the reviewer and we think that this idea is included in the introduction: "A gene is defined as essential if its presence in a genome is required for growth, which is dependent on the growth conditions (9)" A reference has been included to support that point.

We agree with the suggestion that this might impact on the interpretation of the peptidoglycan biosynthesis mutants, we consider that further work is required to be more conclusive. However, the fact remains, in these conditions, genes are not essential. In other organisms, most of the TraDIS/tn-seq assays are performed in lab conditions and all PG genes are categorized as essentials.

2) The manuscript on the whole was rather heavy on bioinformatics and light on experimentation. With all of these wonderful new mutants available, it seems a shame that the only experiment conducted was the demonstration of growth / viability of a couple of strains in a gene set already suspected of not being essential for laboratory growth. Bigger questions could have been easily addressed, such as if these genes are not essential for growth / replication under optimal growth conditions, what about under non-optimal (and likely more physiologically relevant) growth / survival conditions?

We thank the reviewer for this comment. This work is a starting point not just for us but also for the community working on *Planctomyces*, as highlighted by the referee. Indeed, bigger questions will be addressed in subsequent work.

3) the use of optical density rather than a more accurate measure for bacterial growth (cfu counts) is a notable weakness; many things affect OD600 values that have nothing to do with cell division and/or bacterial numbers

We now performed light microscopy and morphological measurements of the mutant cells to address that point. The different mutant cells display similar sizes and shapes, making OD600 values reliable.

Minor comments:

- please add line numbers for subsequent manuscript drafts as it aide with review

We apologize for this mistake; line numbers were included.

- please add additional detail as far as the methods by which the transposon library was prepared. The authors need to clarify that colonies were individually picked during construction, passaged separately, and then then pooled later or whether plates containing colonies were simply swabbed and pooled together at the time of harvesting. This is important, as the existence of a large number of

metabolic mutants could be explained by their uptake of primary and secondary metabolites from other living / dead bacteria on the initial agar plate

Plates were swabbed and pooled together. This detail has been clarified in the the Material and Methods section of the manuscript.

- the authors bring up the potential for compensatory effects of paralogs encoded in the genome; this could be easily demonstrated by assessing the loss of enzymatic / transport activity in the transposon mutants in question, perhaps a few choice examples

We really appreciate the suggestion; however, this would be kept for subsequent work. In addition, please keep in mind the limitations to the number of markers available for Planctomycetes.

“More broadly, our data reveal that the mere presence of the gene is not enough to infer a canonical functional pathway.”

- this is an overly broad interpretation of the data presented. The authors have not demonstrated that any canonical pathway is not functional. They merely observe that some do not appear to be essential based off of their transposon saturation library. They also do not demonstrate (either experimentally or citing the primary literature) functionality of any of the original gene products (something that has proven critical for the exploration of similar pathways in other bacterial systems, such as the Chlamydiales).

In agreement with a similar comment of the reviewer #1, we have modified this statement accordingly.

- The "Homology with Chlamydiae" section does not appear to add much to the authors' story. It is well known that translation, ribosomes, and biogenesis genes are essential in *C. trachomatis*, so it is unclear what 'revealing potential targets to control this important human pathogen' the authors are referring to. Additionally, all PG biosynthesis genes examined to date (including MreB, FtsI, and PBP2) are essential for *C. trachomatis* growth in HeLa / HeP2 cells. One of the more interesting similarities between the two organisms is that both appear to divide via a mechanism akin to budding, but while the process appears to be asymmetric in *P. limnophila*, it is actually a symmetric process in *Chlamydia* tied directly to septal peptidoglycan formation (Liechti 2016, Abdelrahman 2016). This is likely a key defining feature between these two bacterial groups that the authors may wish to highlight. If peptidoglycan biosynthesis is not important for septal formation in *P. limnophila*, what else might be providing the selection to maintain so many PG / 'cell division' genes in the genome?

We have removed this section from the manuscript.

Reviewers' Comments:

Reviewer #1:

Remarks to the Author:

The authors have completely addressed my concerns.

Reviewer #3:

Remarks to the Author:

The authors have made a good faith effort to address the majority of my concerns, particularly those related to overstatements and overly-broad conclusions. The authors have also conducted some additional characterization of their cell division protein mutants. While I still think additional characterization of individual mutant strains would have enhanced the manuscript significantly, I appreciate that the bacterial system is new and future work will expand on these initial observations.

My only remaining concern involves the (now clarified) means of transposon library preparation.

The authors state that the original selection plates containing the initial transposon mutants were pooled to generate the library and that the library was sequenced from DNA extracted from seven samples of that pooled library. If this is accurate, then the authors do not address two potential confounding variables that could impact their results: 1) the potential that their sequencing results represent DNA not only from living microbes but also those that died upon the initial selection and 2) the potential that many of the transposon mutants were only capable of limited / conditional growth on the initial selection plates.

Both of these possibilities (as well as the less likely existence of aneuploidy in the organism) would result in false positives for non-essentiality. Both of these variables could have been minimized somewhat if the original selection plates were replica plated prior to harvest. Absent that, these points should be directly addressed in the manuscript, in either the methods or discussion sections.

Dear Dr Rivas-Marin,

Your manuscript entitled "Essential gene complement of the non-model bacterium *Planctomycetes Planctopirus limnophila* reveals the genetic components of divergent biology" has now been seen again by our referees, whose comments appear below. In light of their advice I am delighted to say that we are happy, in principle, to publish a suitably revised version in Nature Communications under the open access CC BY license (Creative Commons Attribution 4.0 International License).

We therefore invite you to revise your paper one last time to address the remaining concerns of our reviewers and our editorial requests in the attached document(s). At the same time we ask that you edit your manuscript to comply with our policies and formatting requirements and to maximise the accessibility and therefore the impact of your work.

Reviewer #1 (Remarks to the Author):

The authors have completely addressed my concerns.

We thank the reviewer for the revision of our manuscript and for the comments that has improved the quality of our manuscript.

Reviewer #3 (Remarks to the Author):

The authors have made a good faith effort to address the majority of my concerns, particularly those related to overstatements and overly-broad conclusions. The authors have also conducted some additional characterization of their cell division protein mutants. While I still think additional characterization of individual mutant strains would have enhanced the manuscript significantly, I appreciate that the bacterial system is new and future work will expand on these initial observations.

My only remaining concern involves the (now clarified) means of transposon library preparation.

The authors state that the original selection plates containing the initial transposon mutants were pooled to generate the library and that the library was sequenced from DNA extracted from seven samples of that pooled library. If this is accurate, then the authors do not address two potential confounding variables that could impact their results: 1) the potential that their sequencing results represent DNA not only from living microbes but also those that died upon the initial selection and 2) the potential that many of the transposon mutants were only capable of limited / conditional growth on the initial selection plates.

Both of these possibilities (as well as the less likely existence of aneuploidy in the organism) would result in false positives for non-essentiality. Both of these variables could have been minimized somewhat if the original selection plates were replica plated prior to harvest. Absent that, these points should be directly addressed in the manuscript, in either the methods or discussion sections.

We thank the reviewer for his/her suggestions that has improved the manuscript.

We think that the possibility of having DNA from death material is quite low however we have addressed this point in the Methods section. Regarding the aneuploidy we have tested this possibility before in the lab and we can exclude it.